# Quantifying influence of human choice on the automated detection of *Drosophila* behavior by a supervised machine learning algorithm

**Xubo Leng**[1,2☯], **Margot Wohl**[1,3☯], **Kenichi Ishii**[1], **Pavan Nayak**[1], **Kenta Asahina**[1]*

**1** Salk Institute for Biological Studies, La Jolla, California, United States of America, **2** Department of Electrical and Computer Engineering, University of California, San Diego, La Jolla, California, United States of America, **3** Neuroscience Graduate Program, University of California, San Diego, La Jolla, California, United States of America

☯ These authors contributed equally to this work.
* kasahina@salk.edu

**Data Availability Statement:** The original movie files, associated output files of FlyTracker and JAABA, JAABA classifiers, and 'featureConfigEyrun. xml' files are available from the UC San Diego

## Abstract

Automated quantification of behavior is increasingly prevalent in neuroscience research. Human judgments can influence machine-learning-based behavior classification at multiple steps in the process, for both supervised and unsupervised approaches. Such steps include the design of the algorithm for machine learning, the methods used for animal tracking, the choice of training images, and the benchmarking of classification outcomes. However, how these design choices contribute to the interpretation of automated behavioral classifications has not been extensively characterized. Here, we quantify the effects of experimenter choices on the outputs of automated classifiers of *Drosophila* social behaviors. *Drosophila* behaviors contain a considerable degree of variability, which was reflected in the confidence levels associated with both human and computer classifications. We found that a diversity of sex combinations and tracking features was important for robust performance of the automated classifiers. In particular, features concerning the relative position of flies contained useful information for training a machine-learning algorithm. These observations shed light on the importance of human influence on tracking algorithms, the selection of training images, and the quality of annotated sample images used to benchmark the performance of a classifier (the 'ground truth'). Evaluation of these factors is necessary for researchers to accurately interpret behavioral data quantified by a machine-learning algorithm and to further improve automated classifications.

## Introduction

Behavior is the ultimate output of the nervous system [1, 2]. Accurate and quantitative measurements of behavior are vital for evaluating the effects of genetic, neuronal, pharmacological, or environmental perturbations on animals. Traditionally, measurement of behaviors performed by freely moving animals has relied on human observations. Recent advances in computational approaches have transformed this process by replacing human observations with automated computational processes that parameterize animal motions in a high-dimensional space. This information can be then used to classify specific actions through either a

Library Digital Collections (doi: 10.6075/ J0QF8RDZ).

**Funding:** K.I. was supported by the Naito Foundation and the Japan Society for the Promotion of Science Postdoctoral Fellowship Abroad. M.W. was supported by the Mary K. Chapman Foundation and the Rose Hills Foundation. This work was supported by NIH NIGMS R35 GM119844 to K.A. K.A. is a recipient of the Helen McLoraine Development Chair of Neurobiology at the Salk Institute.

**Competing interests:** The authors have declared that no competing interests exist.

supervised or an unsupervised machine-learning algorithm (reviewed in [1, 3–7]. The obvious strengths of automated behavioral classification are the enormous data-processing capacity and the reproducibility of the results. A computer can apply the exact same criteria to every image file, in theory eliminating the variability that may exist within a human observer or among multiple observers. Moreover, an unsupervised machine-learning algorithm may identify new types of behavior that have escaped human attention.

However, computational measurements of behavior invite an inevitable question: how should we evaluate the performance of automated classification? The importance of this question is sometimes overlooked because it is rather trivial if the classification task is unambiguously binary. For example, a face recognition task answers a binary question ('is this person A or not A?'). Most behavioral classification tasks implicitly assume that the answer is likewise binary. However, comparison of the behavioral classifications among multiple observers reveals a considerable level of discrepancy [8–12] that challenges this assumption. This inter-observer variability is often used to promote the superiority of computer-based classification over human observation [4, 6]. However, the performance of every machine-learning-based algorithm must be benchmarked against a "ground truth", which is the annotation by human observers [4]. This means that the human selection of training and ground truth images inherently impact the performance of a computer-based classifier [9, 13]; however, these factors have rarely been assessed systematically. The challenges become more significant as the scale of behavioral data continues to expand. The amount of actual behavioral data one observer can evaluate imposes limitations on the quality of the ground truth data used for performance evaluation and (especially for supervised learning) on the number of training images, which need to be sufficiently diverse to create a reasonably generalizable classifier [9, 14]. It is therefore important to quantitatively assess the relationship between human factors and computational measurements under a variety of situations, especially for behaviors that are variably annotated by human observers.

In this study, we aim to understand how factors controlled by humans, both during training and in the evaluation process, influence the performance of computer classifiers for animal behaviors (Fig 1A). To this end, we first quantified the variability of human observations of three types of social behavior exhibited by pairs of fruit flies (*Drosophila melanogaster*) in multiple sex combinations. In parallel, we developed a series of supervised automated classifiers for these behaviors using a collection of training movies and then quantitatively compared the results of human and computer classification of another dataset. Our results show that the probability that a given behavior bout is detected as a particular behavior by the classifier correlates with the aggregated confidence levels of the human annotators. The performance of the classifiers improved as the diversity of the training files increased, mainly by reducing misclassification of types of behavior that were only present in a subset of movies. Each of the motion-related features curated by the creator of the tracking program assumed different levels of importance for each classifier; features concerning the relative position of the two flies helped improve the classification accuracy for social behaviors. These results suggest that the variability of human observations in fact reflects the variability inherent in animal behaviors, which can be quantified objectively by the confidence levels of well-trained automated classifiers. However, the noticeable impact of training file diversity on classifier performance indicates that it is vital for classifier creators to disclose the nature of the training files before applying the classifier to novel experimental paradigms.

## Materials and methods

### Experimental animals

The complete genotypes of *Drosophila* are listed in Table 1 and S1 Table. *Tk-GAL4[1]* [15], P1[a] split GAL4 [16] (*R15A01-p65AD:Zp* (in attP40) (RRID:BDSC_68837); *R71G01-Zp:GAL4DBD*

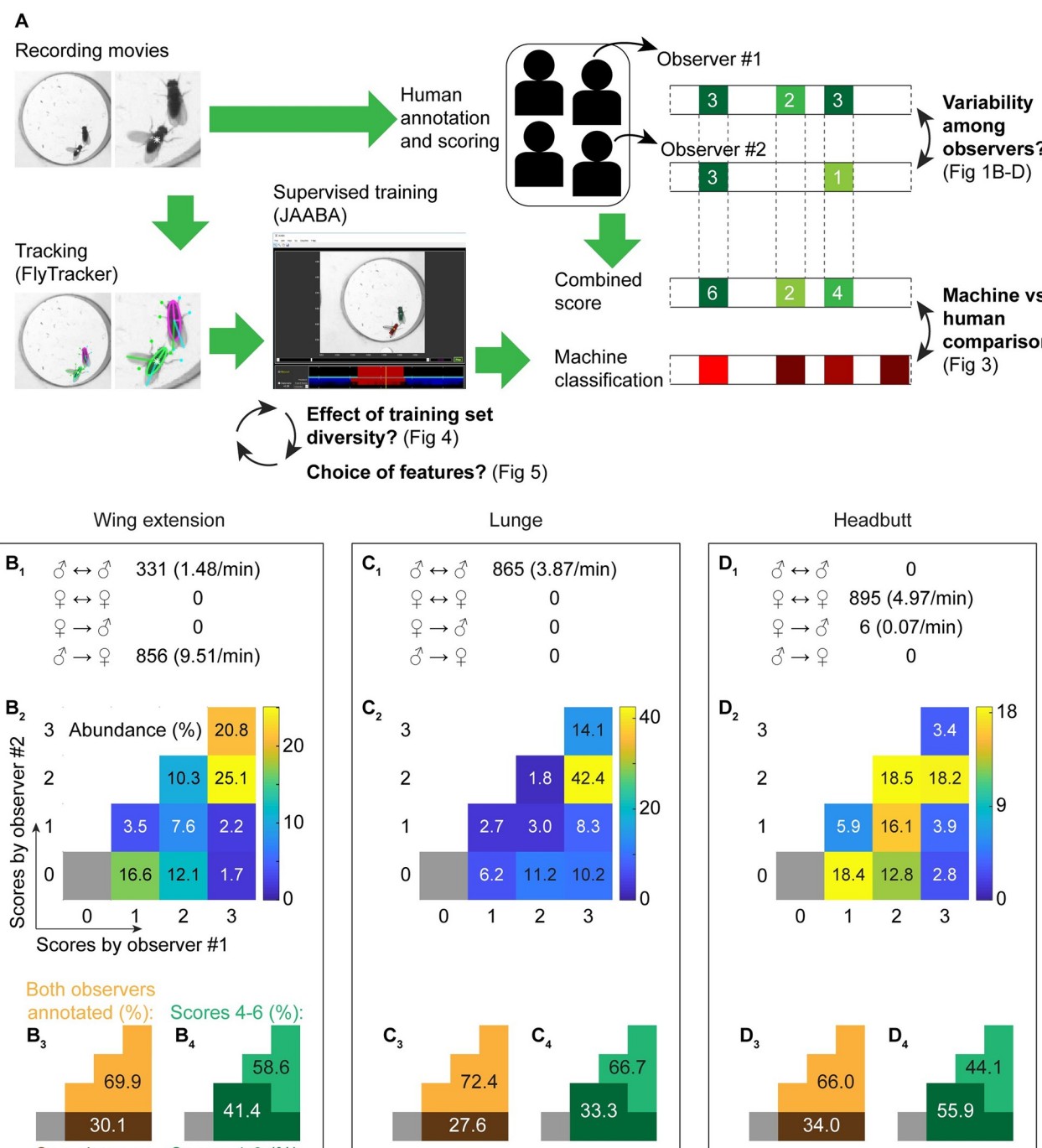

**Fig 1. Variability of human annotation of *Drosophila* social behaviors.** (A). Schematic of workflow and evaluations performed in this study. Movies of a pair of *Drosophila* adults were annotated both by human observers and by machine-learning-based automated classifiers. Inter-observer variability was quantified (B-D) and the performance of human and machine annotations were subsequently compared (Fig 3). The effects of the diversity of training movies (Fig 4) and features (Fig 5) were also quantified. (B-D). Summary of human annotations for wing-extension (B), lunge (C), and headbutt (D) behaviors. The total number of annotated behavioral bouts and frequency are categorized according to interaction type in ($B_1$-$D_1$). The distributions of human score combinations are shown in 4-by-4 grids with pseudocolor representing relative abundance (scale bars on the right of each grid) ($B_2$-$D_2$), and are also broken down according to whether bouts were counted by one or two observers ($B_3$-$D_3$) and by combined score ($B_4$-$D_4$).

**Table 1. Complete descriptions of the movies annotated by human observers.**

| Movie names | Length | Well # used for training | Genotypes of flies | Genotypes of target flies (wing clipped), if different | Sex combination |
|---|---|---|---|---|---|
| 033015_NPF3-CsChrimsonattP40MG-2.avi | 8 min. 30 sec. | 1, 2, 12 | *w*/Y; *20XUAS-CsChrimson:mVenus* in attP40/+; *NPF-GAL4*/+ | | ♂ vs. ♂ |
| 041815_1_m_g_Otd-FLPoChrimsonmvenusattP2_f2a20.avi | 8 min. 30 sec. | 4, 9 | *w, Tk-GAL4¹*/Y; *Otd-nls:FLPo* in attP40/+; *20XUAS>stop>CsChrimson:mVenus* in attP2/+ | | ♂ vs. ♂ |
| 042015_12_m_g_Otd-FLPo Chrimsonmvenusattp2_f10a10.avi | 8 min. 30 sec. | 3, 10, 11 | *w, Tk-GAL4¹*/Y; *Otd-nls:FLPo* in attP40/+; *20XUAS>stop>CsChrimson:mVenus* in attP2/+ | | ♂ vs. ♂ |
| 042415_4_m_g_Otd-FLPoChrimson TdTomattP2_CsHeis_F2a20.avi | 8 min. 30 sec. | 1, 4, 9 | *w, Tk-GAL4¹*/Y; *Otd-nls:FLPo* in attP40/+; *20XUAS>myr:TopHAT2>CsChrimson:tdTomato* in attP2/+ | Wild-type (Canton-S) ♂ | ♂ vs. ♂ |
| 042815_assay1.avi | 30 min. | 6, 7, 9, 12 | Wild-type (Canton-S) ♂ | | ♂ vs. ♂ |
| 042815_assay4.avi | 30 min. | 1, 5, 7 | Wild-type (Canton-S) ♂ | Wild-type (Canton-S) ♀ | ♂ vs. ♀ |
| 050815_assay9.avi | 30 min. | 9, 11 | Wild-type (Canton-S) ♀ | Wild-type (Canton-S) ♀ | ♀ vs. ♀ |
| 082615_CSMH_SF.avi | 30 min. | 1, 3, 5, 7 | Wild-type (Canton-S) ♀ | Wild-type (Canton-S) ♀ | ♀ vs. ♀ |
| 100815_4.avi | 10 min. | 1, 4, 10, 12 | *w, Tk-GAL4¹*/*w*; *Otd-nls:FLPo* in attP40/+; *20XUAS>myr:TopHAT2>CsChrimson:tdTomato* in attP2, *fru⁴⁻⁴⁰*/*fruᴹ* | *+*/Y; *+*; *fruᶠ*/*fru⁴⁻⁴⁰* | Other (fruM ♀ vs fruF ♂) |
| 2016_02_15_CsMH_M_SH2.avi | 5 min. | 1 | Wild-type (Canton-S) ♂ | Wild-type (Canton-S) ♂ | ♂ vs. ♂ |
| 2016_02_15_CsMH_M_SH3.avi | 5 min. | 2 | Wild-type (Canton-S) ♂ | Wild-type (Canton-S) ♂ | ♂ vs. ♂ |

- Well # was counted from left to right, top to bottom.
- A fly that carries 2 X chromosomes and *fruᴹ*/*fru⁴⁻⁴⁰* alleles at the *fru* locus is called "fruM female". A fly that carries 1 X chromosome and 1 Y chromosome, and *fruᶠ*/*fru⁴⁻⁴⁰* alleles at the *fru* locus, is called "fruF male". These nomenclatures are adapted from [22, 24].

(in attP2) (RRID:BDSC_69507)), *Otd-nls:FLP* [15], *UAS-dTRPA1* [17], and *20XUAS-FRT-dSTOP-FRT-CsChrimson:mVenus* (in attP2) [18] are gifts from David Anderson (California Institute of Technology). *20XUAS-CsChrimson:tdTomato* (in VK00022) and *20XUAS-FRT-myr:TopHAT2-FRT-CsChrimson:tdTomato* (in VK00022 or in attP2) were created by Barret Pfeiffer in the lab of Gerald Rubin (HHMI Janelia Research Campus) and kindly shared by David Anderson. *NP2631* [19] is a gift from Daisuke Yamamoto (Tohoku University). *fruᴹ*, (RRID:BDSC_66874) and *fruᶠ* (RRID:BDSC_66873) [20] are gifts from Barry Dickson (HHMI Janelia Research Campus). *dsxᶠᴸᴾ* [21] is a gift from Stephen Goodwin (University of Oxford). *TRH-GAL4* (RRID:BDSC_ 38389) is a gift from Matthew Kayser (University of Pennsylvania). *NPF-GAL4* (RRID:BDSC_25682), *TRH-GAL4* (RRID:BDSC_38388), *20XUAS-CsChrimson:mVenus* (in attP40) (RRID:BDSC_55135), and *fru⁴⁻⁴⁰* (RRID:BDSC_66692) were obtained from Bloomington *Drosophila* Resource Center in the University of Indiana.

All flies were collected as virgins and were maintained at 25˚C, 60% relative humidity. Virgin males and females were reared in a vial with standard *Drosophila* cornmeal media for 6 days, except flies that carry *dsxᶠᴸᴾ* (see S1 Table for details), which were reared for 14 days. For optogenetic experiments, the tester flies were reared on food containing 0.2 mM all-*trans* retinal (MilliporeSigma, Cat#R2500, 20 mM stock solution prepared in 95% ethanol), and vials were covered with aluminum foil to shield light. Mated wild-type (Canton-S) females were prepared by allowing wild-type males to mate with 4-day old virgin females for 2 days. Flies were transferred to vials containing fresh food media every 3 days until the day of the experiment.

## Behavioral assays

Behavioral assays were conducted as described elsewhere [22], although the optogenetic stimulation paradigms differed between movies. Thermogenetic neuronal manipulations were conducted as described in [15]. Briefly, vials that contain testing flies were incubated in a water bath at 28°C for 20 minutes, and were transferred to a behavioral arena. Recording was started 5 minutes after the transfer to allow flies to acclimate. All movies were recorded in .AVI format at 60fps.

As explained in more detail in the following section, recorded fly pairs were separated into "training" pairs and "evaluation" pairs. Training pairs were used for developing JAABA-based behavior-specific classifiers, while evaluation pairs were used for both behavior annotations by human observers and for the evaluation of JAABA classifier performances in comparison to human annotations. A fly pair used for one purpose was never used for another. The separation of training and evaluation pairs is necessary to prevent "over-fitting" of the learning algorithm.

See S1 Table for details of experimental conditions for each movie. All movies (in .avi format) are available as described in Data Availability Statement.

## Tracking of flies

Acquired movies were first processed by the FlyTracker package version 1.0.5 [23] (http://www.vision.caltech.edu/Tools/FlyTracker/), which runs on MATLAB (The Mathworks, Inc.). The regions of interest were manually defined as circles or rectangles that correspond to the chamber of each arena. Foreground and body thresholds were adjusted for each movie for optimal segmentation of body and wing across at least 30 random frames. Note that segmentation is prone to error when two flies are in proximity or overlapping, or a fly is climbing a wall. Some of these cases were discussed in Result section. All tracking parameters can be found in the 'calibration.mat' file associated with each movie.

The identities of flies were confirmed for the following cases: 1) a male-female pair, in which case the sex was identified by the body size and morphology of the posterior end of the abdomen, 2) a male-male pair in which a wing of one of the flies was clipped for identification. The switching of two flies ('identity swap') was manually corrected using the 'Identity correction' function of the "visualizer" program in the FlyTracker package.

FlyTracker output files for all movies used in this study are available as described in Data Availability Statement.

## Human behavioral annotations

Each observer annotated wing extensions, lunges, and headbutts using the "Behavior annotation" function of the "visualizer" program. An observer first determined start and end frames for a given behavioral bout. Then, an observer used the "Certainty" section to specify how confident the annotation for each bout was with three levels: 'maybe' (1), 'probably' (2), and 'definitely' (3). This annotation process created a behavior-specific data structure in which a human confidence score of 0 (no label), 1, 2, or 3 is given to each frame for each fly of every evaluation pair. A single confidence score is assigned to a bout. In rare cases in which the fly of interest performed a behavior continuously with a changing degree of perceived certainty, an observer split the given bout into multiple "bouts", and gave different values of confidence to each bout. While all observers used all three confidence levels for all three annotated behaviors, the relative frequency of use of these confidence levels was discretionary to each observer. Prior to annotating, observers watched select frames of training pair movies together to form a general consensus on target behaviors. Each observer completed annotation independent of each other. See S2–S4 Tables for the complete information of human annotations used in this study.

A movie of an evaluation pair was annotated by two observers. To create combined human annotated bouts, Bout-based combined human annotation was created by merging overlapping annotations via the union operation. For any given bout, start or end frames annotated by the two observers may be shifted, due to subjective judgement regarding which frame the start or end of the behavior is called. For wing extensions, we sometimes found cases in which two annotators segmented bouts differently, and a bout defined by one observer was divided into 2 or more bouts by another observer. These cases were considered as separate bouts, even though the union operation creates one combined human annotation in such cases. We declared that a bout is annotated by two observers if the bouts from the two observers overlapped for one frame (17 ms) or more. For a bout that was annotated by one observer, start and end frames recorded by the observer become the combined human annotation.

The confidence scores given by the two observers to a bout were summed to produce the human combined score for this bout (resulting in a 1-to-6 confidence scale). When more than one bout annotated by the first observer corresponded to one bout annotated by the second observer, we recorded the human combined score for each of the separate bouts. As is detailed below, the highest confidence score among the multiple bouts was used as a representative score of the first observer when a single human combined score is needed for this type of situation.

Frame-based combined human annotation was created by calculating the sum of confidence scores by the two observers for every frame.

## Training of automated classifiers

Frame-by-frame classifiers for wing extensions, lunges, and headbutts were created using the machine learning algorithm JAABA [9, 22, 24]. As stated above, we only used training pairs for classifier development. For each movie, a JAABA folder was created after the identity correction was complete. An .xml file that allows the FlyTracker output to be read in the JAABA platform ('featureConfigEyrun.xml') and JAABA folders with output files for all movies used in this study are available as described in Data Availability Statement.

Details of all training frames for or against each behavior for all training pair movies are available in S1 Table. These frames were accumulated through iterative improvements of classifiers. First, a few dozen bouts of clear behaviors and a similar number of obvious non-behavior frames were labeled as the true behavior and "none", respectively. After initial training, training pairs were classified, and obvious sources of false positives were marked as "none", while a behavior of interest which did not receive high confidence values by the interim classifier was labeled as a "true" behavior. These processes were repeated until we did not observe noticeable improvements of performance, at which point the classifier was considered fully trained. This definition is operational for this study, and it is not meant that the "fully trained" classifier was expected to perform under any experimental conditions or for any genotypes.

Classifiers trained by a subset of training pairs were generated by first removing training frame labels from the fully trained classifier for the given behavior. The classifier was then re-trained anew using only the frames from the specific type of training pairs. Classifiers trained without specific features or rules were generated from the fully trained classifiers in the following steps, except for classifiers trained without relative features. First, features or rules were removed by unchecking the target features or rules in the "Select Features" function in JAABA. The classifier was then re-trained using the same training frames and settings, and saved as a new classifier.

A classifier that did not use relative features was created in the following steps. First, a new JAABA project, in which relative features were removed *a priori* using the "Choose perframe

features" function at the opening window, was created. Movies that contain all training pairs along with all training labels were then imported from the corresponding fully trained classifier using the "Import Exp and Labels from Jab" function. The classifier was then re-trained, and saved as a new classifier.

For downsampling of the training frames, the number of training frames for each fly, for each label ('behavior' or 'non-behavior'), was reduced according to the downsampling ratio by randomly choosing the training frames. The precise downsampling rate slightly deviates from the labeled rate in figure panels, as downsampling seldom generates integer frame numbers, and at least one frame was chosen from each fly for each label regardless of the downsampling rate. Ten independent downsampling and training iterations were applied for each downsampling rate to calculate the average and 95% confidence intervals of precision, false positive rates, and recall.

All classifiers used in this study is available as described in Data Availability Statement.

## Comparison of human annotation and JAABA classification

JAABA classifies a behavior by returning a confidence score for each frame. For bout-based evaluation, we first defined a JAABA bout as a series of continuous frames that has a JAABA confidence score higher than the threshold value. Since this is a frame-by-frame classification, fragmentation of a seemingly single behavioral bout can happen when a bout is relatively short (often the case for lunges or headbutts), or at the edge of a JAABA-defined bout. As was discussed in the main text, we applied a maximum gap filling filter and minimum bout length filter to smooth the fragmented JAABA bouts. We first converted the non-JAABA bout frames (with a JAABA score below the threshold) that are sandwiched by frames with a JAABA score higher than the threshold to JAABA bout frames, while keeping the JAABA score intact. Then, we searched for bouts that had a duration of or above the minimum bout length of choice. After this smoothing process, we recorded the start and end frames of each of the corrected JAABA bouts. The average JAABA score for a given JAABA bout was calculated as (sum of frame-by-frame JAABA score within the JAABA bout)/(total number of JAABA bout frames).

To compare human annotation and JAABA classification on a per-bout-basis, we matched JAABA bouts to combined human annotated bouts using the following procedure. A combined human annotated bout was declared "matched" if a JAABA bout overlapped with the combined human annotated bout for one or more frames (over 17ms). Such a human annotated bout is called a true positive bout. In reality, human annotated bouts and JAABA bouts do not always match one-to-one. For bout-based evaluation, a human annotated bout was used as a reference. Specifically, when one human annotated bout overlapped with multiple JAABA bouts, the JAABA bouts were collectively counted as 1 matched bout. When more than one human annotated bout overlapped with a single JAABA bout, the number of human annotated bouts was the number of matched bouts. To put it the other way, a true positive bout is a category based upon a combined human annotated bout, and not on a JAABA bout.

A false negative bout was defined as a human annotated bout that did not match to any JAABA bout. A false positive bout was defined as a JAABA bout that did not match to any combined human annotation. For a false negative bout, we created a virtual JAABA bout, with length equal to the length of the human annotation intersection. The average JAABA score for these false negative bouts were calculated for the duration of this virtual JAABA bout. For a false positive bout, we created a virtual "human annotated" bout with its human combined score of 0. The average JAABA score for combined human annotated bouts was first calculated using all frames in matching JAABA bouts, which was then associated to a corresponding combined human annotated bout. We chose to do this because (1) it is difficult to split a

combined human annotation when an annotation by one observer overlapped with multiple annotations by another observer, and (2) this operation is consistent with our calculation of the average JAABA score for false positive bouts. In the case (1), we used the highest combined human score as the representative value for the bout. This means that the total number of bouts present in these figure panels are slightly less than the total number of the combined human annotated bouts used elsewhere (such as for calculation of precision and recall). When a single combined human annotated bout was matched by multiple JAABA bouts, the average JAABA score was calculated as an average of all the matched JAABA bouts. When multiple combined human annotated bouts were matched by a single JAABA bout, the single average JAABA score was assigned to each of the matched combined human annotated bouts.

A precision rate represents the ratio of true positive bouts among all positive bouts (true or false), and was calculated as follows:

$$\frac{number\ of\ positive\ bouts}{(number\ of\ true\ positive\ bouts) + (number\ of\ false\ positive\ bouts)}$$

A recall rate represents the ratio of true positive bouts among the total human annotated bouts, and was calculated as follows:

$$\frac{number\ of\ true\ positive\ bouts}{(number\ of\ true\ positive\ bouts) + (number\ of\ false\ negative\ bouts)}$$

For frame-based comparison, each frame is declared "matched" if the frame has simultaneously 1) a JAABA score that is above threshold, and 2) a combined human score of 1 or higher. A frame that has a combined human score of 1 or higher, but a JAABA score below threshold is declared false negative, and a frame that has an above-threshold JAABA score but no human annotation is declared a false positive. Precision and recall rates were calculated in the same manner as for bout-based comparison.

## Generation of shuffled dataset

To address whether the perceived correlation between human combined scores and the average JAABA scores could be observed by chance, we created a series of shuffled data sets from our experimental data set by randomizing a confidence score between 1 and 6 for each human annotated bout, at the same time preserving the percentage of each score category as in the original data. The false positive bouts were kept as false positives. We created 50 such shuffled data sets to evaluate the parameter distributions for statistical analyses.

## Statistical analysis

The details of results of all statistical tests are shown in S1 Datasets. Kruskal-Wallis one-way ANOVA test was used to address whether distributions of average JAABA scores for human annotated bouts are correlated with combined human scores. When the p-value was below the critical value of 0.01, the pairwise 2-sided Mann-Whitney U-test was used to ask whether the median of average JAABA scores for bouts with neighboring combined human scores was significantly different. For this test, we combined both true positive bouts and false negative bouts that had the given combined human scores. Bonferroni correction was applied to the critical p-value for multiple comparisons.

For comparison of shuffled data sets and the experimental data set, the null hypothesis was that there was no correlation between human combined scores and the average JAABA score for a given human annotated bout. If the null hypothesis was correct, we would expect that 1) the observed distribution of average JAABA scores for human annotated bouts would be

within a variation reasonably expected by the randomized data set, and 2) the observed recall rate for each human combined score would fall within the range of fluctuations reasonably expected from the randomized data sets. To test the first possibility, we performed the Kruskal-Wallis test for each of 50 shuffled data sets. To test the second possibility, we calculated "recall" rates for bouts that belong to each combined human score in each shuffled data set. We then calculated the 95% confidence intervals of the "recall" rates for shuffled data sets, for each combined human score, and asked whether the observed recall rate was within the interval.

A violin plot (created by Bastian Bechtold; https://github.com/bastibe/Violinplot-Matlab) was used to represent relative abundance of the number of JAABA bouts or frames that have each of the 7 human combined scores (including false positives, which has a score of 0). For a combined human annotation that had multiple overlaps, the highest human combined score among those separate bouts was taken as the representative human combined score. Violins were created separately for true positives and false negatives. The width of a violins represents the kernel density estimate of the JAABA score statistics for all bouts or frames within the violin. Subsequently, the width of violins for a given behavior was scaled by the ratio relative to the category with the largest number of bouts or frames. Violins for training bouts or frames were created separately, and their width were adjusted according to the relative abundance between positive and negative training bouts or frames.

Ninety-five percent confidence intervals for shuffled datasets and classifiers with down-sampled training frames were calculated using $t$ distribution as follows:

$$m \pm t_{0.025}(n-1) \times \frac{s}{\sqrt{n}}$$

Where n is the sample number (10 in this case), $m$ is the sample mean, $s$ is the standard error, and $t_{0.025}$(n-1) is the upper 0.05/2 = 0.025 critical value for the $t$-distribution with n-1 degrees of freedom (9 in this case).

## Results

### Consistency and variability of human classifications

We first wished to quantify the variability in animal-behavior classifications made by human observers. To this end, we recorded interactions between a number of *Drosophila* pairs and had two trained observers independently annotate the behaviors. Both male and female flies show a variety of stereotypical actions in the context of social interactions. We focused on three types of actions: (1) unilateral wing extensions (henceforth referred to as wing extensions), which are an important part of male-type courtship behavior toward a female [25, 26], (2) lunges, which are a major component of attacks in inter-male aggressive behavior [27, 28], and (3) headbutts, which are a major component of attacks in inter-female aggressive behavior [29, 30]. These three types of actions were chosen because they are frequently observed in specific combinations of the sexes and their motions are relatively unambiguous. We asked the observers to report not only the occurrence of these behaviors but also their subjective confidence level for each annotation, from 1 (least confident) to 3 (most confident). These graded annotations allowed us to quantify how conspicuous the given behavior appeared to human observers.

In all, movies of 30 pairs of flies with a total length of ~534 minutes were independently annotated by two human observers, who were assigned from a pool of four trained scientists (see Table 1 and S1–S4 Tables for details). Each of the three behaviors was observed primarily for a specific combination of sexes, consistent with previous reports. Wing extensions were

performed primarily by males toward females (a low number of male-to-male wing extensions were also observed; Fig 1B$_1$). Lunges were performed exclusively among pairs of male flies (Fig 1C$_1$), while headbutts were performed predominantly among pairs of female flies (Fig 1D$_1$). Although trained observers generally agreed on classification of behaviors, we found that a noticeable number of bouts were annotated by only one of the two observers. For wing extensions, 58.6% of the total annotated bouts received a combined confidence score of 4–6 from the two observers (Fig 1B$_{2,4}$). However, 30.1% of the bouts were annotated by only one of the two observers (Fig 1B$_{2,3}$). Likewise, 66.7% of the total annotated lunge bouts received a combined score of 4–6 (Fig 1C$_{2,4}$), but 27.6% of the bouts were annotated by only one observer (Fig 1C$_{2,3}$). Lastly, 34.0% of the headbutt bouts were annotated by only one observer (Fig 1D$_3$). The more subtle nature of headbutt motions possibly accounts for the lower proportion of bouts that received a combined score of 4–6 (44.1%) (Fig 1D$_{2,4}$). These data suggest that even the "stereotypical" social behaviors of *Drosophila* contain a certain degree of perceived variability.

Lunges and headbutts are "ballistic" behaviors of short and relatively constant duration (median of 83 ms for lunges (S1A Fig) and 67 ms for headbutts (S1B Fig)). On the other hand, the duration of wing extensions can vary greatly (S1C and S1D Fig). Interestingly, wing extensions of longer duration tended to be scored higher by human observers than wing extensions of shorter duration (S1E Fig), suggesting that bout-based analysis of scores may underestimate the consistency of human annotations. Therefore, we analyzed the scores for wing extensions frame by frame. This analysis revealed that 70.6% of the frames received a combined score of 4–6 and 23.4% of frames were annotated by only one observer (S2 Fig). These numbers suggest that consistency among observers is indeed higher at the frame level than at the bout level. We therefore perform both bout-based and frame-based analyses for wing extensions in the following sections. In general, we found that the difference between bout-based and frame-based analyses was quantitative rather than qualitative.

## Automated classifiers quantitatively reflect the confidence of human judgments

Our observations above illuminate a noticeable degree of variability in the annotations of human observers. Similar variability has been reported when more than one person annotates the same movies of behaving nematodes [8], flies [9, 10], and mice [11, 12]. We wondered how this seemingly variable "ground truth" for animal behaviors would be reflected when benchmarking computer classification. To answer this question, we developed a set of well-trained automated classifiers for the three above-mentioned behaviors [22, 24] using the machine-learning-based platform JAABA [9]. In each frame, parts of the fly body were labeled and parameterized using FlyTracker [23], which computes 13 basic feature values related to fly position and motion. The program then generates the first and second derivatives for each feature, creating 39 features in total that are subsequently utilized by JAABA. Compared with unannotated frames, frames annotated by human observers had distinct z-score distributions (Fig 2A$_1$, 2B$_1$ and 2C$_1$) and variances (which likely result in distinct derivatives) (Fig 2A$_2$, 2B$_2$ and 2C$_2$). This suggests that FlyTracker features contain information that JAABA can use to differentiate annotated frames from non-behavior frames.

We aimed to develop JAABA classifiers that perform robustly. As training with a diverse set of movies is important for developing reliable JAABA classifiers [9, 24], we trained a wing-extension classifier with 78,482 frames (1,308 seconds in total), a lunge classifier with 11,360 frames (189 seconds in total), and a headbutt classifier with 10,351 frames (173 seconds in total) (see S1 Table for a complete description of the training movies). Note that the fly pairs in

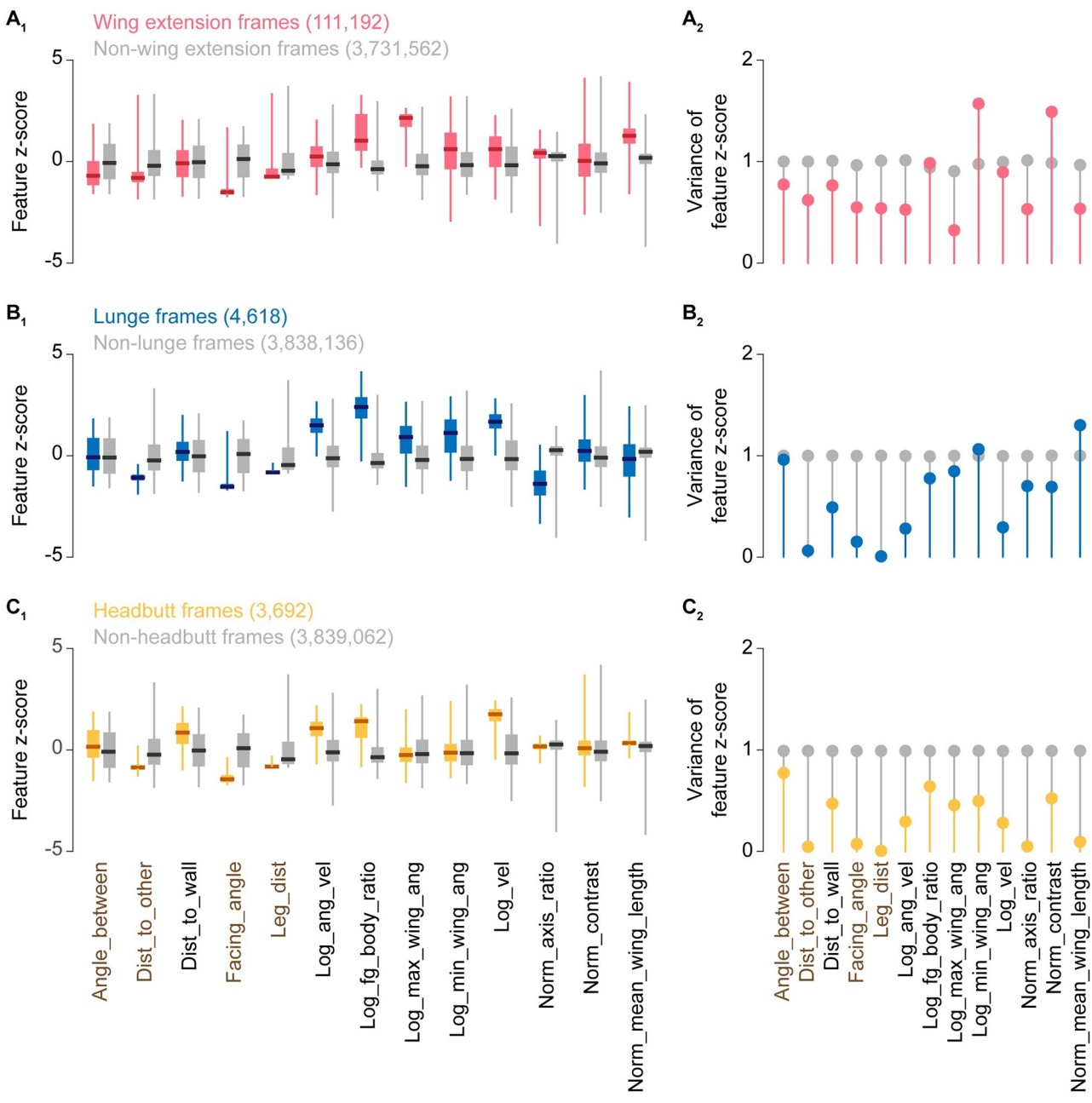

**Fig 2. Frame-based z-score distributions and their variances according to behavior labels.** Distribution of frame-based feature z-scores ($A_1$, $B_1$, $C_1$) and variances ($A_2$, $B_2$, $C_2$) according to human annotations for wing extension (A), lunge (B), and headbutt (C). Z-score distributions are plotted in boxplot, where a thick bar represents median, a box represents 25 and 75 percentiles, and whiskers represents 0.5 and 99.5 percentiles. Features calculated from relative positions of the 2 flies (relative features) are shown in brown.

the training frames were different from the fly pairs in the frames used for testing (i.e., the frames annotated by human observers). We also smoothed raw JAABA-detected bouts by eliminating bouts that were shorter than at least 98% of manually annotated bouts (S1A, S1B and S1D Fig), and by filling short gaps sandwiched by frames that received JAABA scores above the detection threshold value. These filters corrected for fragmentation of behavioral bouts that were sometimes generated by frame-based classification in JAABA (see Materials

and methods for details). We used recall (the ratio of human annotations detected by the classifier to all *human* annotations) and precision (the ratio of human annotations detected by the classifier to all *classifier* annotations) to evaluate the performance of the classifiers. The set of smoothing parameters that resulted in the optimal trade-off between recall and precision with a detection threshold of 0.1 was used for subsequent analyses unless otherwise noted (S3 Fig).

We then quantitatively compared the classification results from human observers and the JAABA classifiers by calculating recall and precision. We found that the JAABA classifiers reliably detected behavioral bouts that received high human confidence scores. For bouts that received a score of 4 or higher (Fig $1B_4$–$1D_4$), recall was 96.0% for the wing-extension classifier, 95.3% for the lunge classifier, and 87.9% for the headbutt classifier. All three JAABA classifiers had almost perfect recall for behavioral bouts that received a score of 6 (Fig 3B, 3E and 3H).

As expected, the detection threshold was positively correlated with precision (Fig 3A, 3D and 3G, S6A Fig) and inversely correlated with recall (Fig 3B, 3E and 3H, S6B Fig) for all three behaviors. Interestingly, recall steadily decreased for all classifiers as the combined confidence scores dropped (Fig 3C, 3F and 3I, S6C Fig). For bouts that received scores of 3 or lower (Fig $1B_4$–$1D_4$), recall fell to 31.5% for wing extensions, 73.7% for lunges, and 64.3% for headbutts. Moreover, the median JAABA score tended to be lower for bouts with lower combined human scores (Fig 3C, 3F and 3I, S6C Fig); this accounts for the lower recall for bouts that received lower confidence scores from human observers. These observations indicate that benchmarking values (such as recall) can be quantitatively influenced by the ground truth annotation of human observers.

The apparent correlation between human confidence levels and JAABA scores is intriguing given that the JAABA classifiers were trained with binary labels ("true" or "false") instead of graded weights. JAABA scores of true and false for the training frames were largely, if not completely, separated (Fig 3C, 3F and 3I, S6C Fig), suggesting that the behaviors included in the training frames were mostly unambiguous. Recall would have no correlation with the combined confidence score if human confidence levels were randomly assigned subjective values that had no relationship with either the other observer's confidence levels or JAABA scores, which are objective "confidence" levels determined by the algorithm. Indeed, a permutation test confirmed that this correlation is highly unlikely to be generated by chance. When we randomized the human confidence scores to the bouts that JAABA detected in a size-matched manner, the probability of an uneven distribution of JAABA scores across human confidence scores was very small ($p < 0.01$ by Kruskal-Wallis test; Fig 3J). The p-values from our experimental data were also many orders of magnitude smaller than the smallest p-values obtained with randomized samples (Fig 3J). In addition, the 95% confidence intervals of the expected average recall for permutated datasets were lower than the observed values for bouts with high combined confidence scores and higher for bouts with low scores (Fig 3K). These results suggest that the confidence level of the human observers can be predicted to a certain extent by the JAABA classifiers.

We also noticed that false-positive bouts for all three behaviors had low JAABA confidence values (Fig 3C, 3F and 3I, S6C Fig), indicating that most false positives barely passed the detection threshold. To address whether these false positives stemmed from misclassification of certain types of motions, we manually inspected all the false-positive bouts (Tables 2–4). Interestingly, we found that a noticeable number of "false positives" appeared similar to true behaviors. These actions likely escaped the observers' attention. For the wing-extension classifier, 22% of "false positive" bouts appeared to be actual wing extensions (Table 2). For the lunge classifier, 25% appeared to be actual lunges or lunge-like motions that were not completed, and another 20% were lunge-like striking actions during high-intensity tussling (Table 3). For the headbutt classifier, 7% appeared to be actual headbutts, and another 20% were ambiguous

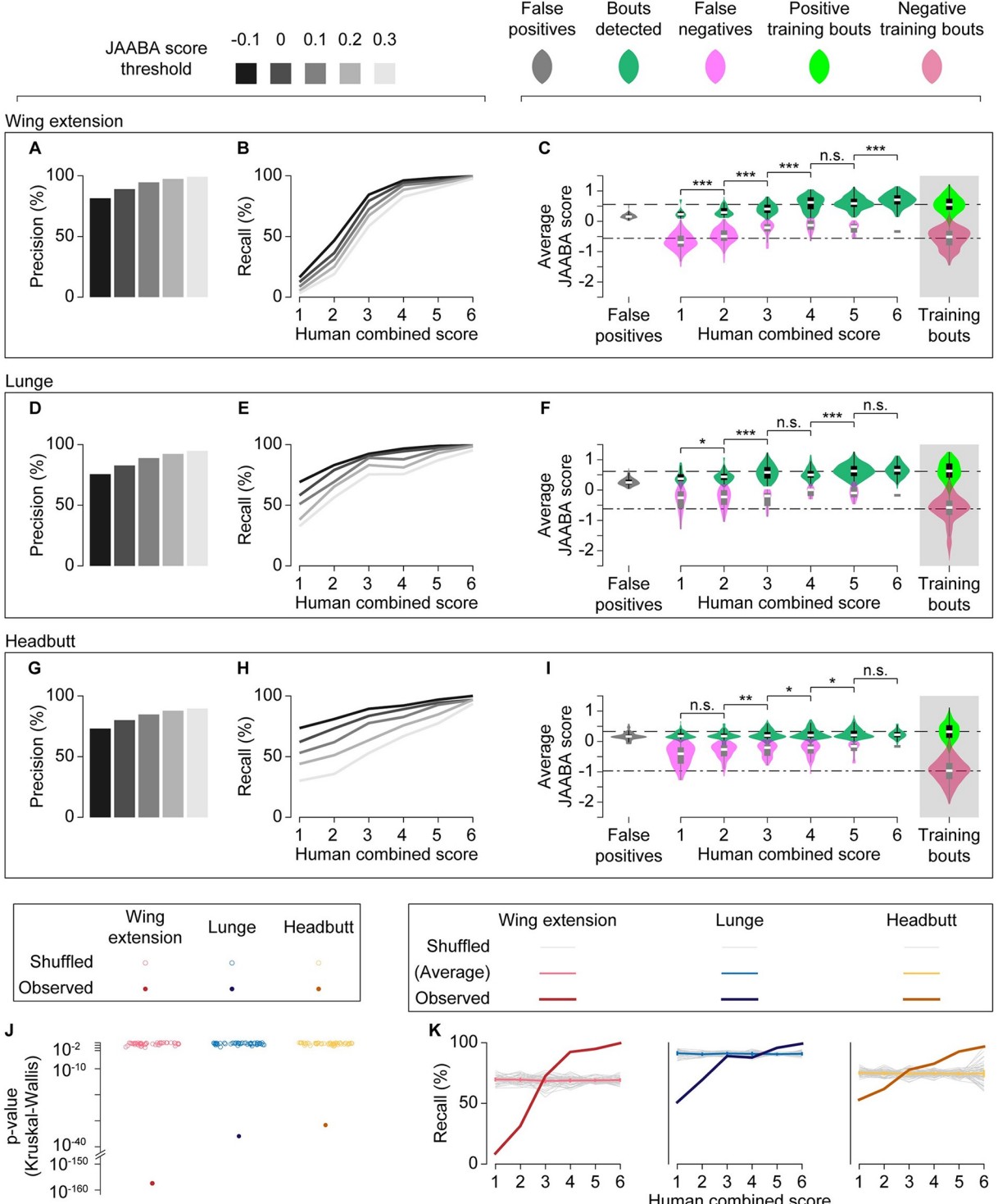

**Fig 3. Human confidence and JAABA confidence are correlated.** (A-I) Precision (A, D, G) and recall (B, E, H) of fully trained classifiers for wing extensions (A, B), lunges (D, E), and headbutts (G, H) are shown for varied JAABA score thresholds, as indicated at the top of the figure. For recall, detected bouts were binned according to human combined scores of 1 to 6. In (C) (wing extensions), (F) (lunges), and (I) (headbutts), the distributions of average JAABA scores for true-positive (green), false-negative (pink), and false-positive (gray) bouts are shown as both violin plots (see Materials and methods for definitions) and box plots. As references, distributions of positive (light green) and negative (crimson) training bouts are shown at right, and the median values for positive and negative training bouts are shown by even and uneven broken lines, respectively. n.s. p > 0.05, * p < 0.05, ** p < 0.01, *** p < 0.001 by Kruskal-Wallis one-way ANOVA and post-hoc Mann-Whitney U-test. (J). Kruskal-Wallis p-value distributions of shuffled (open circles) and observed (filled circles) data sets across human combined scores. (K). Recall rates across human combined scores for shuffled and observed data sets at a JAABA score threshold = 0.1 (observed data sets are replotted from (B), (E), and (H)). Average and 95% confidence intervals for shuffled data are shown in light colors.

**Table 2. Annotations of false positives by the wing extension classifier.**

| Types of motions | False positive counts | | | | |
| --- | --- | --- | --- | --- | --- |
| | ♂ vs. ♂ | ♂ vs. ♀ | ♀ vs. ♀ | Other (fruM ♀ vs fruF ♂) | Total (%) |
| Actual wing extensions | 5 | 0 | 0 | 6 | 11 (22.4) |
| Wing glooming | 6 | 0 | 2 | 4 | 12 (24.5) |
| Incomplete wing closure | 8 | 0 | 1 | 3 | 12 (24.5) |
| Wing tracking error (including actions when flies are on or near the wall) | 7 | 1 | 0 | 1 | 9 (18.4) |
| Wing threats | 2 | 0 | 0 | 1 | 3 (6.1) |
| Others | 1 | 1 | 0 | 0 | 2 (4.1) |

• Green categories indicate motions that are similar to actual behaviors.

**Table 3. Annotations of false positives by the lunge classifier.**

| Types of motions | False positive counts | | | | |
| --- | --- | --- | --- | --- | --- |
| | ♂ vs. ♂ | ♂ vs. ♀ | ♀ vs. ♀ | Other (fruM ♀ vs fruF ♂) | Total (%) |
| Actual or incomplete lunges | 24 | 0 | 0 | 1 | 25 (25.3) |
| Lunge-like actions during tussling | 20 | 0 | 0 | 0 | 20 (20.2) |
| Actions when flies are on or near the wall (ex. falling from the wall) | 17 | 1 | | 2 | 20 (20.2) |
| Receiving lunge or headbutt | 13 | 0 | 2 | 0 | 15 (15.2) |
| Fast autonomous motions (jumping, rolling on floor, etc.) | 14 | 1 | 0 | 0 | 15 (15.2) |
| Copulation attempt | 0 | 2 | 0 | 0 | 2 (2.0) |
| Reaction to other fly's fast autonomous motions | 0 | 0 | 2 | 0 | 2 (2.0) |

**Table 4. Annotations of false positives by the headbutt classifier.**

| Types of motions | False positive counts | | | | |
| --- | --- | --- | --- | --- | --- |
| | ♂ vs. ♂ | ♂ vs. ♀ | ♀ vs. ♀ | Other (fruM ♀ vs fruF ♂) | Total (%) |
| Actual headbutt | 0 | 1 | 7 | 1 | 8 (6.6) |
| Jerking toward the other fly | 1 | 0 | 23 | 0 | 24 (19.8) |
| Walking toward the other fly while extending a leg ('reaching') | 0 | 0 | 24 | 0 | 24 (19.8) |
| Receiving lunge or headbutt | 0 | 0 | 22 | 0 | 22 (18.2) |
| Fast autonomous motions (jumping, rolling on floor, etc.) | 0 | 0 | 21 | 0 | 21 (17.4) |
| Reaction to other fly's fast autonomous motions | 1 | 0 | 7 | 1 | 9 (7.4) |
| Pushing the other fly | 0 | 0 | 8 | 0 | 8 (6.6) |
| Actions when flies are on or near the wall (ex. falling from the wall) | 1 | 0 | 2 | 0 | 3 (2.5) |
| Lunge | 1 | 0 | 0 | 0 | 1 (0.8) |
| Wing threat with fast motion toward the other fly ('charge') | 1 | 0 | 0 | 0 | 1 (0.8) |

"jerking" motions which were difficult to clearly distinguish from headbutts (Table 4). Excluding such incidents, the common false positives for the wing-extension classifier involved wing motions, such as grooming (Table 2). For the lunge and headbutt classifiers, short, quick motions (such as when a fly received a lunge or headbutt, or when a fly fell from the wall) were often misclassified (Tables 3 and 4).

These false positives can be at least partially explained by the types of feature-value deviations associated with a given behavior (Fig 2). For instance, wing-extension frames annotated by human observers had high z-scores for maximum wing angle and high variance for minimum wing angle. Annotated lunge and headbutt frames had high z-scores and low variance for velocity. Together, these observations suggest that the source of the false positives was not necessarily random "noise". Although we made rigorous efforts to minimize common types of false positives during training, it proved difficult to eliminate them without sacrificing recall values (Fig 3A, 3B, 3D, 3E, 3G and 3H).

Overall, these quantitative analyses suggest that the perceived variability in *Drosophila* behaviors is not solely due to subjective human artefacts, but is at least partially attributable to variability in the motion of the flies themselves. Deviations from "stereotyped" actions can be represented by lower confidence both in human observers and in automated behavioral classifiers. Inherent variability in animal behaviors that have historically been regarded as "stereotypical" is consistent with an emerging view that animal behaviors can be represented as a probability distribution in a continuous parameter space [6, 7, 31].

## Diversity of training samples affects the robustness of automated classifiers

Whether supervised or unsupervised, automated animal-behavior classifiers are developed with training samples, which are usually chosen by humans. These samples likely cover only a portion of the behavioral repertoire that a given animal species can exhibit. We next addressed how the choice of training movies affected the performance of our classifiers.

A male fly performs wing extensions vigorously toward female flies, and to a lesser extent toward other males (see Fig 1B$_1$). The recall of the wing-extension classifier trained only on movies of male–female pairs (48,992 frames) was comparable to the recall of the fully trained classifier (Fig 4C, red). However, the precision was only 64.5%, noticeably lower than that of the fully trained classifier (Fig 4A, red). The lower precision was largely due to a high false-positive rate with male–male pair movies (Fig 4B, red). Manual inspection revealed that a large number of false positives were fast flicking motions of wings that males often show when paired with another male. Addition of female–female (Fig 4A and 4B, pink) and male–male (Fig 4A and 4B, purple) training pairs steadily improved the precision. In contrast, the recall remained largely unaltered by the increase in the diversity of the training movies (Fig 4C) (See also S6D–S6F Fig for frame-based analyses).

Lunges are performed predominantly among males. When the lunge classifier was trained only on movies of pairs of male flies (7,921 frames in total), its precision was 76.3%, again lower than that of the fully trained classifier (Fig 4D). False-positive bouts from this classifier largely occurred in movies of male–female pairs, while the false-positive rates in movies of male–male pairs remained largely the same (Fig 4E). Manual inspection revealed that the majority of false positives in male–female pair movies came from misclassification of copulation attempts as lunges. The addition of male–female training pairs, which contained only negative training frames (1,927 frames, all labeled as "not lunge"), largely eliminated false positives in this condition (Fig 4D, purple), but also decreased the recall for bouts with relatively low combined human scores (1–4) (Fig 4F, purple). The addition of female–female training pairs (395 frames) had little impact on precision (Fig 4D, green), but the addition of both male–

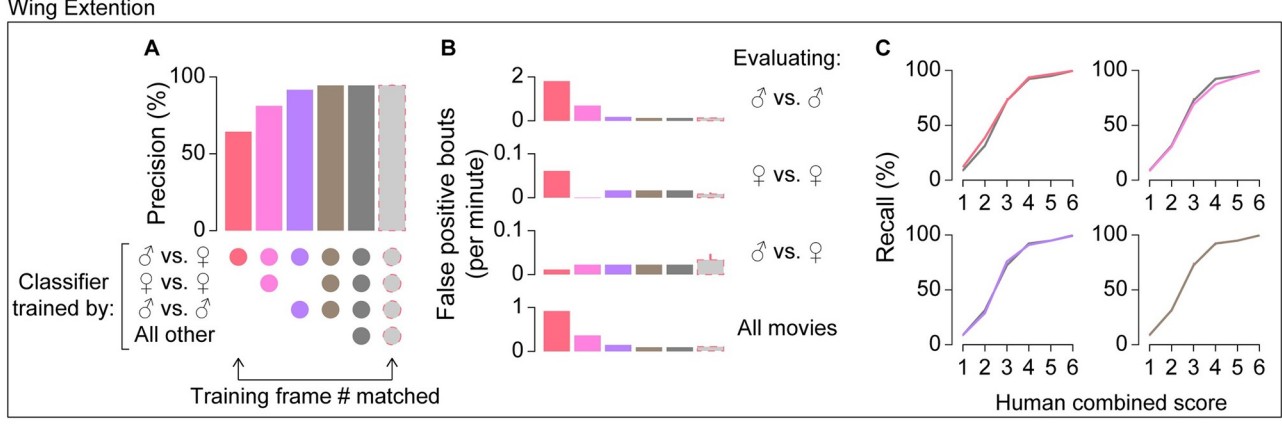

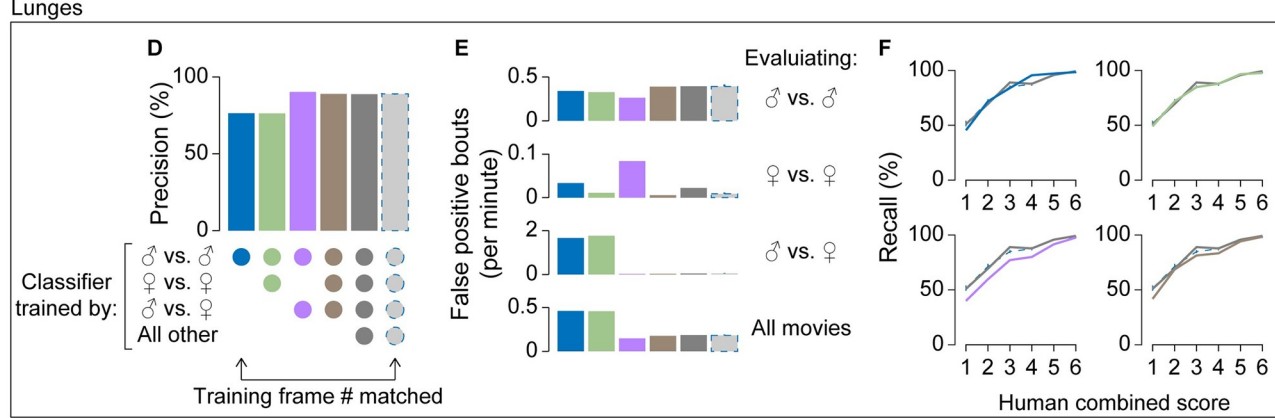

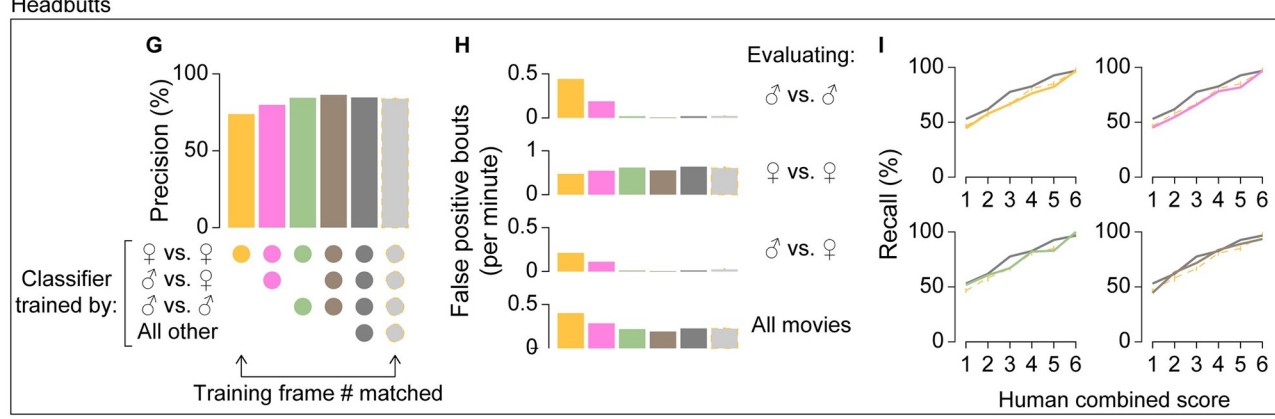

**Fig 4. Classifier performance improves as diversity of training frames increases.** Precision (A, D, G), false-positive rates (B, E, H), and recall (C, F, I) of classifiers for wing extensions (A-C), lunges (D-F), and headbutts (G-I) are plotted according to the types of training movies used (indicated by the circles below A, D, G). False positive rates are shown separately for the evaluating movie types indicated on the right. "All other" movies include *fruitless* mutants as indicated in S1 Table. Gray bars with broken outlines (A, B, D, E, G, H) and broken lines (C, F, I) represent the mean and 95% confidence intervals of the classifiers trained with frames downsampled proportional to the ratio of the training frames from a single type of movie (left-most bars on A, D, G) to the entire number of training frames. Note that the 95% confidence intervals are generally very small. Also, recall for wing-extension and lunge classifiers with downsampled training frames are very similar to those for fully trained classifiers. Precision and recall for classifiers trained by "all movies" (shown in gray) are replotted from Fig 3.

female and female–female training pairs improved the precision while limiting the decrease in recall (Fig 4D–4F, brown).

Lastly, for the headbutt classifier, both precision (Fig 4G) and recall (Fig 4I) increased as training movies of male–female pairs and male–male pairs were added. Unlike the wing-extension and lunge classifiers, the largest source of false positives was female–female pairs for all versions of the headbutt classifier trained with an intermediate diversity of training movies (Fig 4H). Nonetheless, the decrease in false positives when male–male and male–female training movies were added helped improve the precision (Fig 4G and 4H).

In the experiments above, more frames were added to the training set as the variety of training frames increased, raising the possibility that it was the increased number of training frames that led to the improvements in classifier performance. To address this, we trained classifiers on fewer training frames, while maintaining the training-frame diversity. We did this by randomly selecting training frames across all training movies, with the ratio of frames proportional to those used with the fully trained classifier (we call this process "downsampling" of the training frames: see S5 Table for details of classifiers with downsampled training frames). We found that a downsampling rate of 5% for the lunge classifier (S4D and S4E Fig), and 25% for the wing-extension (S4A, S4B, S6G and S6E Figs) and headbutt (S4G and S4H Fig) classifiers, was sufficient to achieve precision comparable to the fully trained counterparts. Male–female pairs accounted for 62.3% of the entire training frames for the fully trained wing-extension classifier, male–male pairs accounted for 69.7% of the entire training frames for the lunge classifier, and female–female pairs accounted for 51.7% of the entire training frames for the headbutt classifier. When we adjusted the downsampling rate to these values, the precision of all classifiers was, predictably, similar to the precision of the fully trained classifiers (Fig 4A, 4D and 4G). Moreover, their 95% confidence intervals were well above those of classifiers trained on a single type of fly pair. The results from the downsampled training indicate that the low precision of classifiers trained on limited types of training movies is not simply due to the reduction in the number of training frames relative to the fully trained classifiers.

For wing extension and lunges, classifiers trained on downsampled training frames and fully trained classifiers showed largely comparable recall (Fig 4C and 4F, S6F Fig). By contrast, the recall of the fully trained headbutt classifier was sometimes well above the 95% confidence intervals for recall of the downsampled headbutt classifiers (Fig 4I). This could mean that the current fully trained headbutt classifier may be further improved by adding more training frames. It is also possible that the fully trained classifier might be over-fitted to the evaluation movies.

These results demonstrate that selecting a variety of training frames is critical for improving the robustness of the behavior classifiers, even if some of the movies do not contain the behavior of interest. Interestingly, classifiers trained solely with a specific type of movie appear comparable to the fully trained classifiers only when their performance was evaluated with the same type of movie (male–male pair movies for lunges, male–female pair movies for wing extensions, female–female pairs for headbutts). This suggests that a seemingly well-performing behavior classifier that is validated only for a specific combination of sexes may not perform well with other combinations.

## Robustness of classifiers when tracking information is incomplete

In machine vision approaches that estimates an animal's pose [13, 23, 32, 33], animal posture is subdivided into a combination of parameters (features) that describe the position of each body segment (such as head, limbs, etc.). As previously mentioned, JAABA creates a series of rules that use feature and feature derivatives as inputs for its machine-learning algorithm [9]. Importantly, these features and rules are defined by the creator of a program. We next

examined whether a small number of features can contain sufficient information for JAABA-based behavior classifications, or whether a diversity of features and rules is collectively important for robust performance of a classifier.

Differences in the z-score distributions and variance of particular features between annotated and unannotated frames (Fig 2) suggest that classifiers may utilize these information-rich features to distinguish behavior frames. Indeed, we found that each classifier used rules derived from each feature with different weights (Table 5). We re-trained classifiers after removing 1 or 3 of the most highly weighted features, and compared their precision and recall with those of the fully trained classifiers. The precision of these classifiers was noticeably lower than the precision of the fully trained classifiers. In particular, the bout-based precision of the wing-extension classifier without its most weighted feature, maximum wing angle, was only 44%, and removal of the 3 most-weighted features further decreased the precision to 22% (Fig 5A, dark purple; see also S6J Fig for frame-based statistics). The precision of the lunge and head-butt classifiers showed qualitatively similar, but less exaggerated, trends (Fig 5D and 5G, dark purple). By contrast, removal of the 3 least-weighted rules from the training process did not noticeably affect either precision (Fig 5A, 5D and 5G, light gray bars) or recall (Fig 5C, 5F and 5I, broken gray lines). Interestingly, classifiers lacking the most-weighted features retained the tendency to detect bouts with higher human confidence scores better than bouts with lower human confidence scores (Fig 5C, 5F and 5I). Overall, these observations suggest that key features have a large impact on the reliable detection of a behavior.

We wondered whether these highly weighted features alone contain sufficient information to create reliable behavior classifiers. To test this possibility, we used only the three most highly weighted features when training the classifiers. Wing-extension classifiers trained this way performed surprisingly well, showing precision and recall very similar to the fully trained classifier (Fig 5A–5C, light purple). By contrast, the lunge and headbutt classifiers that were trained this way had 24% and 19% lower precision, respectively, than the corresponding fully trained classifiers (Fig 5D–5I, light purple). Together, these results show that information relevant for behavior classification can be distributed across many features, even though some features contribute more than others to the performance of a classifier. This underscores the value of using a variety of features when developing a reliable classifier.

**Table 5. Weights given to features in each classifier (ascending order).**

| Wing extension classifier | | Lunge classifier | | Headbutt classifier | |
|---|---|---|---|---|---|
| Feature | Weight | Feature | Weight | Feature | Weight |
| log_max_wing_ang | 5.678 | norm_axis_ratio | 3.416 | log_vel | 3.569 |
| facing_angle | 2.690 | dist_to_other | 1.948 | dist_to_other | 2.298 |
| norm_mean_wing_length | 2.249 | norm_mean_wing_length | 1.870 | facing_angle | 1.230 |
| log_min_wing_ang | 1.152 | facing_angle | 1.616 | log_ang_vel | 1.165 |
| dist_to_wall | 1.118 | log_max_wing_ang | 0.937 | dist_to_wall | 1.119 |
| norm_axis_ratio | 0.867 | leg_dist | 0.899 | leg_dist | 1.049 |
| log_fg_body_ratio | 0.661 | dist_to_wall | 0.697 | norm_mean_wing_length | 0.960 |
| leg_dist | 0.650 | log_vel | 0.625 | log_max_wing_ang | 0.853 |
| log_vel | 0.498 | angle_between | 0.539 | log_fg_body_ratio | 0.842 |
| dist_to_other | 0.342 | log_min_wing_ang | 0.459 | norm_axis_ratio | 0.816 |
| log_ang_vel | 0.228 | log_fg_body_ratio | 0.411 | norm_contrast | 0.774 |
| norm_contrast | 0.133 | norm_contrast | 0.406 | log_min_wing_ang | 0.380 |
| angle_between | 0.110 | log_ang_vel | 0.246 | angle_between | 0.334 |

• Features in brown indicate relative features

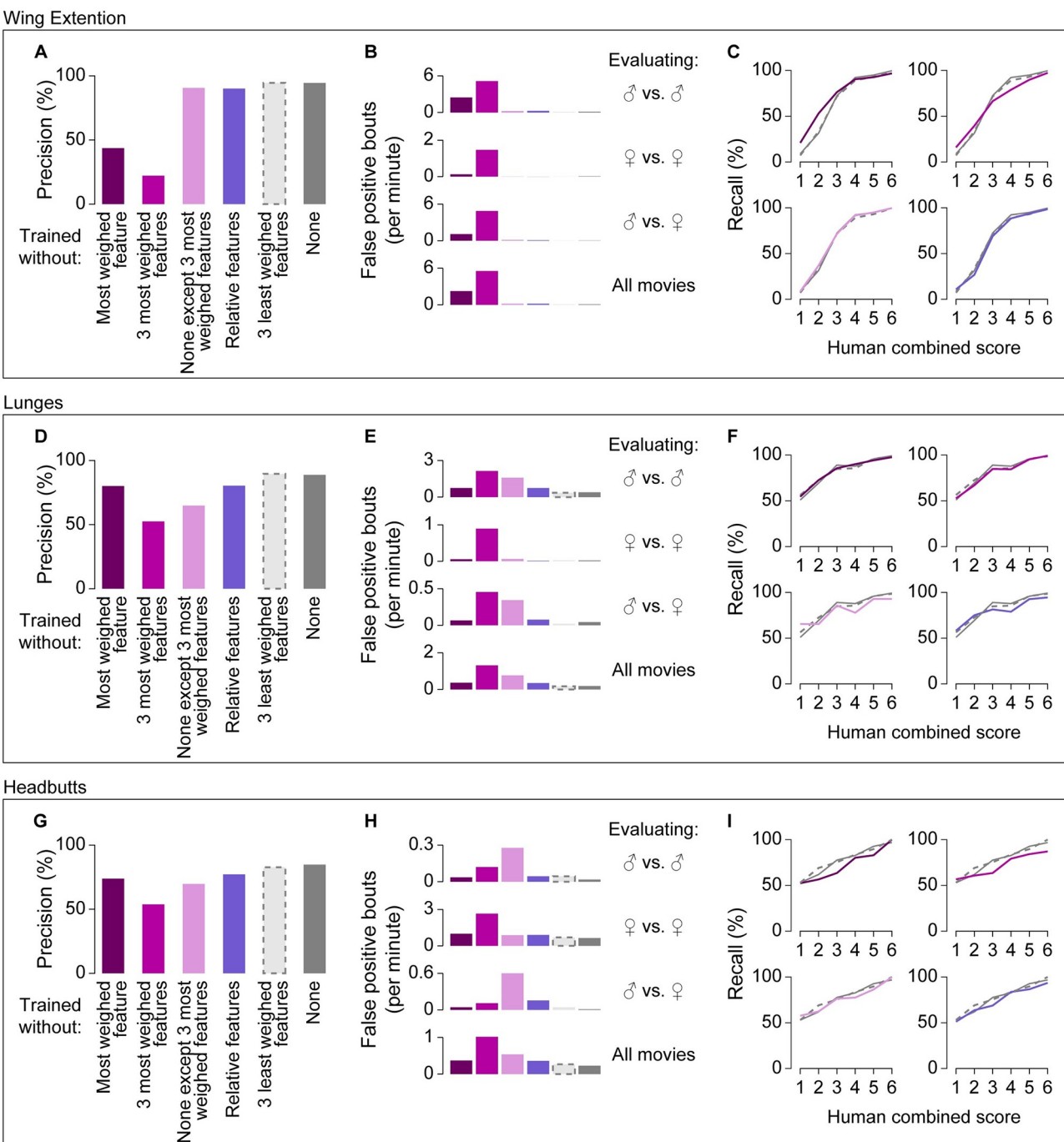

**Fig 5. Performance of classifiers change when some rules or features are removed.** Precision (A, D, G), false-positive rates (B, E, H), and recall (C, F, I) of classifiers for wing extensions (A-C), lunges (D-F), and headbutts (G-I) are plotted according to the features not available for training on JAABA (shown below A, D, G). False-positive rates are shown separately for the evaluating movie types indicated on the right. Precision and recall for classifiers trained with all features (shown in gray) are replotted from Fig 3.

JAABA rules further increase the diversity of information that can be useful for behavior classification. A list of the most-weighted rules for each classifier (S6 Table) indicates that all three classifiers distribute weights to rules derived from a relatively large number of features, suggesting that aggregates of diverse rules can collectively encode information relevant for

behavioral classification. Indeed, when we disabled usage of the 3, 5, or 10 most-weighted rules for fully trained classifiers, both precision (S5A, S5B, S5D, S5E, S5G and S5H Fig) and recall (S5C, S5F and S5I Fig) degraded only marginally. Upon removal of these rules, other rules that derived from corresponding features often took over as the most-weighted rules (S6 Table), suggesting that rules derived from the same feature may contain redundant information for reliable behavioral classification.

Under natural conditions, social behaviors are expressed largely toward other animals and, to a certain extent, are defined in relation to the other animals. This raises the possibility that the accurate classification of a social behavior may require information about the relative positions of the participating animals, such as the distance between the two individuals, the orientation angle, and so on. We tested this possibility by training the classifier on the same set of training frames but without using the features that concern the relative position of the two flies ("relative features"). Among the 13 basic features, 9 features were defined solely by the autonomous properties of a single fly, whereas the remaining 4 were relative features (Fig 2) [23]. At least one relative feature was among the 3 most-weighted features for all three fully trained classifiers (Table 4). We found that both the precision and recall of the classifiers trained without relative features decreased modestly. The decrease in bout-based precision was 4.3% for the wing-extension classifier, 8.5% for the lunge classifier, and 7.5% for the headbutt classifier (Fig 5A, 5D and 5G, S6J Fig, dark blue), and was due to an increase in false-positive rates across movie types (Fig 5B, 5E and 5H, S6K Fig, dark blue). Bout-based recall across all human score categories was −1.8% for the wing-extension classifier, and −2.7% for the lunge and headbutt classifiers (Fig 5C, 5F and 5I, dark blue). These results indicate that classifier accuracy can benefit from information about the relative position of the flies.

## Discussion

With technological advances in recording and movement tracking, many types of automated approaches have been applied to quantify animal behaviors (reviewed in [6, 7, 34]). However, even automated approaches involve human choices and judgments, from the design of the algorithm to the choice of training and benchmarking samples. Relatively few systematic efforts have been made to quantitatively evaluate the impact of human factors at each step, or to reconcile automated results with results obtained from traditional human observation. Here, we systematically compared the annotation of *Drosophila* social behaviors by a group of human observers and by JAABA-based automated classifiers under a variety of perturbations. Our results lead to three important conclusions. First, the variability of human annotations is correlated with the "confidence" levels of automated classifiers, suggesting that the animal behaviors themselves are the source of at least part of the variability. Second, the use of diverse training samples is crucial to ensure robust performance of automated classifiers. Lastly, information about the relative position of experimental animals can improve the accuracy of automated classifiers.

Some classification tasks can assume an unambiguous ground truth. For instance, the goal of a human face-recognition task is binary: does a given face belong to the target person or not? Behavioral classification can involve more nuanced situations because a degree of variability may exist even in what is traditionally regarded as "stereotypical behavior" [7]. Animals can abort ongoing behavior before completion [35], or can show variability in the execution of a learning-dependent motor program during [36, 37] or even after [38] training. We found that human observers gave different levels of confidence in their classification of *Drosophila* social behaviors. It is possible to argue that such inter-observer variability would disappear with perfectly trained observers. However, inconsistencies among human observers have been

repeatedly noted across the behavior classification of multiple species [8–12]. In the absence of a clear definition of a "perfectly trained observer", these variable human classifications need to be accepted as the "ground truth" when evaluating the performance of automated classifiers [4]. It is therefore practically important to assess how the results of automated behavior classification should be benchmarked against a variable "ground truth". We found that the level of human confidence and JAABA confidence values correlated when behavioral bouts were aggregated. Although we tried to avoid using frames with perceived ambiguous behaviors for both positive and negative training), it is possible that some mislabeling of training frames could have caused ambiguous classification [9]. However, because of the diversity of training frames used in this study, we favor an alternative possibility: that the behaviors performed by pairs of flies contain inherent variability. This conclusion implies that it may not be possible to define the ground truth in a binary fashion. If a behavior–even one originally defined by human observers–is distributed across confidence-level space, the boundaries defining the behavior are ultimately decided by human judgment, much like a critical p-value in statistics.

At the same time, our observations reinforce the idea that "subjective" human confidence levels can reflect statistically defined discriminability [39], and argue against the notion that variability in human judgments is simply randomly generated noise. This is consistent with our finding that "high-confidence" behavioral bouts are indeed clearly identified by both humans and automated classifiers. As an operational variable, nomenclature for stereotypical behaviors [25, 28] remains useful for characterizing the nature of behavior and, importantly, necessary to bridge human observations from previous studies with data obtained through automated systems. This is particularly relevant for unsupervised behavioral classification, because the resulting clusters of "behaviors" are named by a human observer [31, 40], sometimes without clear descriptions of the nature of the variability within the given cluster. For specific purposes, eliminating low-confidence behavioral bouts from analyses may be justified [9, 41]. However, these bouts may not be equivalent to non-behavior frames. Instead, they may have biological consequences in the context of social interactions. One possible solution is to weigh behavioral bouts according to the confidence value each bout receives. This approach is analogous to the quantification by scoring for other biological phenomena, such as dye-based feeding amount [42, 43] or aberrations of neuronal morphologies [44, 45].

Diverse training images are important for developing a robust automated classifier because machine-learning algorithms generally assume that the parameter space is the same for the training samples and the testing samples [9, 46]. However, how automated classifiers perform on *types* of movies that are not a part of the training images has not previously been quantitatively analyzed. As expected, we found that classifiers trained with movies that contained only one combination of the sexes had lower precision than the fully trained classifier. Interestingly, the precision and recall of these partially trained classifiers were comparable to the values of the fully trained classifiers when the training and evaluation movies were of the same type. While these results are not surprising in light of the nature of machine-learning algorithms, they illuminate a source of misinterpretation that may go unnoticed. An experimenter may be satisfied with the performance of an automated classifier on the basis of limited evaluation examples and overlook the errors that the classifier may commit for types of movies that the experimenter did not include in either the training or evaluation processes. This situation is analogous to gender- and race-dependent classification bias by a face-recognition algorithm that was not trained on a diverse dataset of faces. The bias became apparent only after diverse faces were used for benchmarking [14]. Since we do not know *a priori* the parameter space of behavior under different genetic and environmental manipulations, one possible interim solution is to

fully disclose the nature of the training and evaluation images so that a new user can be aware of possible limitations for generalization. In the long run, a common depository of behavioral movies taken from a variety of genotypes under diverse conditions, much like a large dataset for annotating objects [47] and human motions [48], may help in the development of a universally applicable behavioral classifier.

Accurate tracking of animals is essential for successful automated classification [34]. The tracking program we used in this study segments fly body parts in two-dimensional images [23]. This means that the tracking becomes inaccurate when the fly changes body orientation along the z-axis. In fact, a certain portion of false positives in all three classifiers were detected when a fly was on the wall, which violates the tracking program's assumptions about the appearance of the body. One solution is to force an animal to pose in largely expected ways in a spatially restrictive arena [31, 32, 49], but it is possible that such environmental constraints can put artificial limitations on animal behaviors. Multiple cameras with different angles [50], or a depth-sensitive camera [51, 52], enable the three-dimensional posture of animals to be visualized and provides more comprehensive information about animal posture.

Each of our classifiers used different features with different weights, suggesting that certain features are critical for accurate detection of behaviors [46]. At the same time, our finding that even features and rules with relatively low weights contribute to improving the performance of classifiers is noteworthy. One interesting question is how multiple pose-estimation packages [13, 33] will perform for automated behavior classification and how animal behavioral classification systems built on pose estimation compare against pixel-based behavioral classifications that do not assume specific animal postures [31, 52].

We also found that the automated classifiers for social behaviors give considerable weight to information related to the relative positions of the animals involved. Elimination of these relative features modestly but noticeably deteriorates the performance of classifiers, suggesting that these features are valuable for accurate behavior classification. Consistent with our observation, a recent report shows that relative features distribute distinctively between different behaviors and states during male courtship behavior to females [53]. For a machine-vision system that tracks animals, calculation of relative features can become computationally expensive as the number of tracked animals increases. As interest in detecting and quantifying social behavior grows, it may become necessary to identify a set of relative features that 1) has high discriminability [53] and 2) a low computational burden [46].

Beyond standardizing behavioral quantification in a reproducible manner, automated behavioral classification methods have the potential to reveal motion patterns and behavioral dynamics that may escape human attention. It is important to note that it is ultimately human intuition and judgment that allows interpretation of results from often multidimensional automated classification. Just like carefully curated expert annotations of select genomic regions greatly facilitated automated annotation of the entire genome, detailed annotations of example animal movies through synthesis of multiple experts' observations will be foundational to ensuring that automated behavioral classification is as informative and objective as possible.

## Supporting information

**S1 Fig. Bout duration distributions for each behavior.** (A-D). Histograms of bout duration for lunge (A), headbutt (B), and wing extension (C, D). Green is the distribution of human annotated bouts, and gray is the distribution of JAABA bouts. For lunges and headbutts, bout duration was binned for every 17 ms (duration of one frame). For (C), bins are indicated below the plot. (D) is the magnified histogram for durations between 0 and 250 ms. Red shades

show the durations that are eliminated by the minimum bout length filter (see text and S3 Fig).
(E). Distribution of wing extension bout durations according to human combined score. Bins
are color-coded as shown on the right.
(TIF)

**S2 Fig. Frame-based statistics of human-annotated wing extensions.** Frame-based summary
of human annotations for wing extension ($A_1$), categorized according to interaction types. Distribution of human score combinations are shown in 4-by-4 grid with pseudocolor ($A_2$) that
represent relative abundance (scale bars on the right of the grid), and the breakdown according
to whether frames were counted by one or two observers ($A_3$) and combined scores ($A_4$) are
also shown. Source data is identical to that used in Fig 1B.
(TIF)

**S3 Fig. Choice of filter parameters for JAABA bout smoothing.** Recall-precision plot of fully
trained classifiers for wing extension (A), lunge (B), and headbutt (C) when average JAABA
score threshold, minimum bout length, and maximum gap to be filled are varied as indicated
below each plot. Green (arrows and rectangles) indicates the parameter combinations chosen
for fully annotated classifiers. Dotted lines are inverse proportion functions that pass the points
indicated by green arrows. These combinations have the near-maximum recall X precision values for each behavior.
(TIF)

**S4 Fig. Performance of classifiers with downsampled training frames.** Mean and 95% confidence intervals (vertical lines) of precision (A, D, G), false positive rates (B, E, H) and recall (C,
F, I) of classifiers for wing extension (A-C), lunges (D-F), and headbutt (G-I) trained with
downsampled frames (indicated below precision plots and inside recall plots). False positive
rates are shown separately for types of movies classified as indicated on the right. Values for
classifiers trained at 100% downsampling rate (all frames used, shown in gray) are replots
from Fig 3.
(TIF)

**S5 Fig. Classifier performs robustly when most weighted rules are removed.** Precision (A,
D, G), false positive rates (B, E, H) and recall (C, F, I) of classifiers for wing extension (A-C),
lunges (D-F), and headbutt (G-I) trained without the most weighted JAABA rules (indicated
below precision plots). False positive rates are shown separately for classifying movie types as
indicated on the right. Values for classifiers trained with all rules (shown in gray) are replots
from Fig 3.
(TIF)

**S6 Fig. Frame-based analyses of wing extensions classifiers.** (A-C) Frame-based plots for
precision (A), recall (B), and average JAABA score distribution for each human combined
score (C) of the results from the fully trained wing extension classifier (corresponding to Fig
3A–3C). (D-F) Frame-based plots for precision (D), false positive rates (E), and recall (F) of
the results from the wing extension classifiers trained with the specific types of training movies
(shown below precision plots) (corresponding to Fig 4A–4C). (G-I) Frame-based plots for
mean and 95% confidence intervals (vertical lines) of precision (G), false positive rates (H) and
recall (I) of classifiers trained with downsampled frames (rates indicated below precision plots
and inside recall plots). False positive rates are shown separately for classifying movie types as
indicated on the right. Frame-based plots for precision (J), false positive rates (K), and recall
(L) of the results from the wing extension classifiers trained with subsets of features (as indicated below precision plots). (corresponding to Fig 5A–5C). For (D-L), dark gray plots

represent the value of the fully trained classifier (bars in (D), (G), (J) are replots of (A), and lines in (F), (I), (L) are replots of (B), respectively, at JAABA score threshold of 0.1).
(TIF)

**S1 Table. Detailed descriptions of the movies used for human annotation and classifier training.**
(XLSX)

**S2 Table. Human annotations for wing extensions.**
(XLSX)

**S3 Table. Human annotations for lunges.**
(XLSX)

**S4 Table. Human annotations for headbutts.**
(XLSX)

**S5 Table. Details of classifiers with dowmsapled training frames.**
(XLSX)

**S6 Table. Most weighted JAABA rules for classifier.**
(XLSX)

**S1 Datasets.**
(XLSX)

## Acknowledgments

We thank Dr. Pietro Perona and Eyrun Arna Eyjolfsdottir for developing and improving Fly-Tracker during the course of this study, Isabella Stelter for assisting with the evaluation of JAABA classifiers, Dr. Terrence Sejnowski and Jorge Aldana for providing us with computational resources, infrastructure, and technical support, and Drs. Eiman Azim, Daniel Butler, and Saket Navlakha for helpful discussions. Stocks obtained from the Bloomington Drosophila Stock Center (NIH P40OD018537) were used in this study.

## Author Contributions

**Conceptualization:** Kenta Asahina.

**Data curation:** Xubo Leng, Margot Wohl, Kenichi Ishii, Pavan Nayak, Kenta Asahina.

**Formal analysis:** Xubo Leng, Margot Wohl.

**Funding acquisition:** Kenta Asahina.

**Investigation:** Margot Wohl, Kenichi Ishii, Kenta Asahina.

**Methodology:** Xubo Leng, Margot Wohl, Kenta Asahina.

**Project administration:** Kenta Asahina.

**Resources:** Kenta Asahina.

**Software:** Xubo Leng, Margot Wohl.

**Supervision:** Kenta Asahina.

**Validation:** Xubo Leng, Kenta Asahina.

**Visualization:** Xubo Leng, Margot Wohl, Kenta Asahina.

**Writing – original draft:** Xubo Leng, Kenta Asahina.

**Writing – review & editing:** Xubo Leng, Margot Wohl, Kenichi Ishii, Kenta Asahina.

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
