## [Decision Letter · Decision Letter 0]

20 Oct 2020

Quantifying influence of human choice on the automated detection of Drosophila behavior by a supervised machine learning algorithm

PONE-D-20-24775

Dear Dr. Asahina,

We’re pleased to inform you that your manuscript has been judged scientifically suitable for publication and will be formally accepted for publication once it meets all outstanding technical requirements.

Kind regards,

Giorgio F Gilestro, PhD

Academic Editor

PLOS ONE

Reviewers' comments:

Reviewer's Responses to Questions

**Comments to the Author**

1. Is the manuscript technically sound, and do the data support the conclusions?

Reviewer #1: Yes

2. Has the statistical analysis been performed appropriately and rigorously? 

Reviewer #1: Yes

3. Have the authors made all data underlying the findings in their manuscript fully available?

Reviewer #1: Yes

4. Is the manuscript presented in an intelligible fashion and written in standard English?

Reviewer #1: Yes

5. Review Comments to the Author

Reviewer #1: With increasing use of classifiers to identify behaviours, a careful analysis of their limitations and possible biases is important. In this paper the authors apply a widely used classifier (JAABA) to three fly behaviours and compare the results to human annotators. They analyse the agreement between human annotators and the automated classifier, show that human confidence is somewhat predictive of automated classifier performance (behaviours that humans are less certain about are more likely to be misclassified by the automated classifier), and investigate how the classifier performance changes with differences in training data and tracking parameters. The paper is thorough, clearly written, and appropriately analysed.

In my opinion it can be published as is.

The data to reproduce the plots and analysis is included in the supplementary table but the video files used to train the classifiers (and the trained classifiers themselves) are not currently available. If possible, it would be useful to share these as well using something like zenodo.org or a similar repository.

One very minor improvement: the equations for precision and recall could be better formatted as equations rather than text to make them more readable.

6. PLOS authors have the option to publish the peer review history of their article (what does this mean?). If published, this will include your full peer review and any attached files.

Reviewer #1: No

---

## [Editor Report · Acceptance letter]

1 Dec 2020

PONE-D-20-24775 

Quantifying influence of human choice on the automated detection of *Drosophila* behavior by a supervised machine learning algorithm 

Dear Dr. Asahina:

I'm pleased to inform you that your manuscript has been deemed suitable for publication in PLOS ONE. Congratulations! Your manuscript is now with our production department. 

Kind regards, 

on behalf of

Dr. Giorgio F Gilestro 

Academic Editor

PLOS ONE